**Data Availability Statement:** All relevant data are within the manuscript and its Supporting Information files.

**Funding:** This research was funded by Boston Heart Diagnostics, Framingham, MA, St. Francis

# Corona Virus Disease-19 serology, inflammatory markers, hospitalizations, case finding, and aging

Ernst J. Schaefer[1,2]*, Latha Dulipsingh[3,4], Florence Comite[5,6], Jessica Jimison[7], Martin M. Grajower[8], Nathan E. Lebowitz[9], Maxine Lang[1], Andrew S. Geller[1], Margaret R. Diffenderfer[1], Lihong He[1], Gary Breton[10], Michael L. Dansinger[2,10], Ben Saida[11], Chong Yuan[11]

1 Laboratory Division, Boston Heart Diagnostics/Eurofins Scientific Network, Framingham, Massachusetts, United States of America, 2 Department of Medicine, Tufts University School of Medicine, Boston, Massachusetts, United States of America, 3 Division of Diabetes and Endocrinology, Saint Francis Hospital and Medical Center, Trinity Health of New England, Hartford, Connecticut, United States of America, 4 Department of Medicine, University of Connecticut School of Medicine, Farmington, Connecticut, United States of America, 5 Clinical Care, Comite Center for Precision Medicine and Health, New York, New York, United States of America, 6 Department of Internal Medicine, Endocrinology/Metabolism/Diabetes, Lenox Hill Hospital/Northwell, New York, New York, United States of America, 7 Clinical Care, Atkinson Family Practice, Amherst, Massachusetts, United States of America, 8 Department of Medicine (Endocrinology), Albert Einstein College of Medicine, The Bronx, New York, United States of America, 9 Clinical Care, Advanced Cardiology Institute, Fort Lee, New Jersey, United States of America, 10 Clinical Affairs Division, Boston Heart Diagnostics/Eurofins Scientific Network, Framingham, Massachusetts, United States of America, 11 Research Division, Diazyme Laboratories, Inc., Poway, California, United States of America

* eschaefer@bostonheartdx.com

## Abstract

Most deaths from severe acute respiratory syndrome coronavirus-2 (SARS-CoV-2) infection occur in older subjects. We assessed the utility of serum inflammatory markers interleukin-6 (IL-6), C reactive protein (CRP), and ferritin (Roche, Indianapolis, IN), and SARS-CoV-2 immunoglobulin G (IgG), immunoglobulin M (IgM), and neutralizing antibodies (Diazyme, Poway, CA). In controls, non-hospitalized subjects, and hospitalized subjects assessed for SARS-CoV-2 RNA (n = 278), median IgG levels in arbitrary units (AU)/mL were 0.05 in negative subjects, 14.83 in positive outpatients, and 30.61 in positive hospitalized patients (P<0.0001). Neutralizing antibody levels correlated significantly with IgG (r = 0.875; P<0.0001). Having combined values of IL-6 ≥10 pg/mL and CRP ≥10 mg/L occurred in 97.7% of inpatients versus 1.8% of outpatients (odds ratio 3,861, C statistic 0.976, P = 1.00 x $10^{-12}$). Antibody or ferritin levels did not add significantly to predicting hospitalization. Antibody testing in family members and contacts of SARS-CoV-2 RNA positive cases (n = 759) was invaluable for case finding. Persistent IgM levels were associated with chronic COVID-19 symptoms. In 81,624 screened subjects, IgG levels were positive (≥1.0 AU/mL) in 5.21%, while IgM levels were positive in 2.96% of subjects. In positive subjects median IgG levels in AU/mL were 3.14 if <30 years of age, 4.38 if 30–44 years of age, 7.89 if 45–54 years of age, 9.52 if 55–64 years of age, and 10.64 if ≥65 years of age (P = 2.96 x $10^{-38}$). Our data indicate that: 1) combined IL-6 ≥10 pg/mL and CRP ≥10 mg/L identify SARS-CoV-2 positive subjects requiring hospitalization; 2) IgG levels were significantly correlated

Hospital, Trinity Health of New England, Hartford,
Ct, Comite Center for Precision Medicine and
Health, New York, NY, Atkinson Family Practice,
Amherst, MA, Advanced Cardiology Institute, Fort
Lee, NJ, and Diazyme Laboratories, Poway, CA.
The authors were employees of Boston Heart
Diagnostics (EJS, ML, ASG, MRD, LH, GB, MLD,
LH, GB, MLD), Trinity Health of New New England
(LD) Comite Center for Precision Medicine and
Health (FC), Atkinson Family Practice (JJ),
Grajower Medical Practice (MMG), Advanced
Cardiology Institute (NEL), and Diazyme
Laboratories (BS, CY). The funders provided
support in the form of salaries for authors but did
not have any additional role in the study design,
data collection and analysis, decision to publish, or
preparation of the manuscript. The specific roles of
these authors are articulated in the 'author
contributions' section.

**Competing interests:** This research was funded by
Boston Heart Diagnostics, Framingham, MA,
St. Francis Hospital, Trinity Health of New England,
Hartford, Ct, Comite Center for Precision Medicine
and Health, New York, NY, Atkinson Family
Practice, Amherst, MA, Advanced Cardiology
Institute, Fort Lee, NJ, and Diazyme Laboratories,
Poway, CA. The authors were employees of Boston
Heart Diagnostics (EJS, ML, ASG, MRD, LH, GB,
MLD, LH, GB, MLD), Trinity Health of New New
England (LD) Comite Center for Precision Medicine
and Health (FC), Atkinson Family Practice (JJ),
Grajower Medical Practice (MMG), Advanced
Cardiology Institute (NEL), and Diazyme
Laboratories (BS, CY). These commercial affiliation
does not alter adherence to PLOS ONE policies on
sharing data and materials. The conclusions
expressed are solely those of the authors.

with neutralizing antibody levels with a wide range of responses; 3) IgG levels have significant utility for case finding in exposed subjects; 4) persistently elevated IgM levels are associated with chronic symptoms; and 5) IgG levels are significantly higher in positive older subjects than their younger counterparts.

## Introduction

Severe acute respiratory syndrome coronavirus 2 (SARS-CoV-2) is the causative agent of the coronavirus disease 2019 (COVID-19) pandemic. A COVID-19 diagnosis is typically made by reverse transcriptase-polymerase chain reaction (RT-PCR) detection of SARS-CoV-2 RNA in naso-pharyngeal (NP), oro-pharyngeal (OP), nasal swabs, or saliva usually within 10 days of exposure [1–6]. Up to 50% of SARS-CoV-2 positive patients can remain asymptomatic; however, such individuals can spread infections [7, 8]. The average onset of symptoms in symptomatic patients usually occurs within 5 days of exposure (range 2–14 days). Antibody testing has been reported to be useful for documenting exposure and potential immunity, as well as for case finding in family clusters and exposed individuals [9–16]. Moreover, treatment of symptomatic COVID-19 patients with convalescent plasma rich in antibodies or specific monoclonal antibodies may be useful in treating the disease [16–21].

In RT-PCR RNA positive subjects, IgM antibody levels may be detectable within a median time of 5 days (range 3–7 days) of symptom onset and generally disappear over time, while IgG and neutralizing antibodies may be detectable within a median time of 14 days (range 10–18 days) of symptom onset and generally persist for many months [9–15, 22, 23]. Similar results for SARS-CoV-2 antibodies have been obtained with chemiluminescence and enzyme-linked immunoassays [9–15]. Levels of IgG antibodies have been shown to correlate with levels of neutralizing antibodies in serum with some assays, but not with others [22, 23]. Antibody testing with some lateral flow devices may be unreliable [24, 25]. It has been reported by the Centers for Disease Control in the United States that about 80% of the total deaths attributed to SARS-CoV-2 occur in subjects ≥65 years of age, while this group only accounts for about 10% of the total cases [26]. Our goals in the current investigation were to assess: 1) the relationships of inflammatory markers and antibody levels in SARS-CoV-2 positive patients requiring hospitalization, as compared to those in positive patients not requiring hospitalization, in order to develop a risk prediction model; 2) the relationships of IgG and IgM antibody levels with neutralizing antibody levels; 3) the clinical utility of such assays in case finding and symptom prediction; and 4) the effects of age and sex on serum SARS-CoV-2 IgG and IgM antibody levels.

## Materials and methods

### Human subjects

We measured serum interleukin-6 (IL-6), high-sensitivity C reactive protein (hs-CRP), ferritin and SARS-CoV-2 IgG, IgM, and neutralizing antibody levels in 100 SARS-CoV-2 RNA negative control subjects, 129 SARS-CoV-2 RNA positive subjects not requiring hospitalization, and 49 SARS-CoV-2 RNA positive subjects requiring hospitalization (median age 48.9 years; 54.5% female; 85% Caucasian, 10% Hispanic, and 7% African American). These subjects were enrolled in an IRB-approved protocol at St. Francis Hospital, Trinity Health of New England (Hartford, CT, USA). All subjects provided informed written consent.

We also measured SARS-CoV-2 IgG and IgM antibody levels on serum samples obtained from 534 outpatients and selected inpatients (median age 46 years, 51.2% female). These samples were submitted to our laboratory by healthcare providers in Boston, the Bronx, Manhattan, and northern New Jersey. Clinical data on these subjects provided by healthcare providers as well as laboratory information were analyzed as anonymized data. We also assessed data in a similar fashion from samples collected by a healthcare provider from employees at a local meat packing plant in Massachusetts (n = 217). In addition, we measured IgG levels in a total of 150,222 serum samples submitted by healthcare providers to our laboratory for antibody measurements between April 6th, 2020 and December 1st, 2020. This number decreased to 83,153 samples when only the first sample was utilized, and this number further decreased to 81,624 after removing subjects without age or gender information. Their median age was 48.0 years (IQR 30–55), and they were 57.77% female. A subset of 61,126 of these subjects (median age 50.0 years [IQR 35–61]; 58.89% female) also had IgM values measured. We also report data from 39 states with more than 100 results.

This type of research is exempted from requirement for human institutional review board (IRB) approval as per exemption 4, as listed at https://grants.nih.gov/policy/humansubjects. htm and at the open education resource (OER) website for research involving human subjects. This exemption "involves the collection or study of data or specimens if publicly available or recorded such that subjects cannot be identified". We had this designation and our research reviewed by the Advarra Institutional Review Board (Columbia, MD). They determined that "had the request for exempt determination been submitted prior to initiation of research activities, the research would have met the criteria for exemption from institutional review board review under 45 CFR 46.104(d)" and, therefore, ruled that this research did not require IRB approval. Anonymized data and material used for all analyses has been uploaded onto the journal website as requested.

## SARS-CoV-2 viral detection

Detection of SARS-CoV-2 RNA in NP, OP, or nasal swabs was performed by reverse transcriptase-polymerase chain reaction (RT-PCR) using Thermo-Fisher TaqPath COVID-19 Combo kits (Waltham, MA). This assay targets a region in the *N* gene, a region in the spike glycoprotein or *S* gene, and a region in the *ORF1* gene for SARS-CoV-2 RNA detection in swab samples. Positive values are those detected at a cycle threshold values of $\leq$37 cycles. Our modified version of this assay which has received emergency use authorization (EUA) from the Food and Drug Administration (FDA) was performed as previously described [5]. Our assay was found to have 100% concordance in 100 positive and 100 negative samples when compared with another RNA assay from Viracor (Lee's Summit, MO) as previously described [4].

## SARS-CoV-2 IgG and IgM chemiluminescence assays

The assays used were the SARS-CoV-2 IgM (catalog number 130219016M) and SARS-CoV-2 IgG (catalog number 130219015M) chemiluminescence assays obtained from Diazyme Laboratories (Poway, CA) as previously described [10, 14, 15]. The assays use 2 recombinant antigens (full-length SARS-CoV-2 nucleocapsid protein and partial-length glycoprotein spike protein). The prediluted sample, buffer, and magnetic microbeads coated with SARS-CoV-2 recombinant antigens are thoroughly mixed and incubated, forming immune-complexes. The precipitate is separated in a magnetic field and washed before *N*-(4-Aminobutyl)-*N*-ethyl-iso-luminol labeled anti-human IgM or IgG antibodies are added and incubated to form additional complexes. After a second precipitation in a magnetic field and subsequent wash cycles, the Starter 1+2 is added to initiate a chemiluminescent reaction. The light signal is measured

by a photomultiplier as relative light units (RLUs), which are proportional to the concentration of SARS-CoV-2 IgM or IgG present in the sample and are converted to arbitrary units or AU/mL.

The SARS-CoV-2 IgG antibody test did not detect SARS-CoV-2 IgM antibodies, and the SARS-CoV-2 IgM antibody test did not detect SARS-CoV-2 IgG antibodies. For cross reactivity experiments, a total of 143 clinical samples were tested with both antibody assays. These samples were confirmed positive for antibodies for various viruses and bacteria: *influenza virus type A*, *influenza virus type B*, *parainfluenza virus*, *respiratory syncytial virus*, *adenovirus*, *EBV NA* IgG, *EBV VCA* IgM/IgG, *Measles* virus, *CMV* IgM/IgG, *Varicella zoster virus*, *Mycoplasma pneumoniae* IgM/IgG, *Chlamydia pneumoniae* IgM/IgG, *Candida albicans*, *ANA*, *HCoV-HKU1*, *HCoV-OC43*, *HCoV-NL63* and *HCoV-229E*. These experiments were carried out at Diazyme Laboratories. All 143 samples were negative for SARS-CoV-2 IgG/IgM with DZ-Lite SARS-CoV-2 IgG/IgM CLIA kits. In addition, these assays were found to have no cross reactivity with antibodies for non-SARS-CoV-2 coronavirus strains *HKU1*, *NL63*, *OC43*, or *229E*. Multiple serum samples with IgM concentrations ranging from 0.86–10.27 AU/mL and IgG concentrations ranging from 8.04–67.92 AU/mL had 0.1 mg/mL of the S protein and 0.1 mg/mL of the N protein added. After 10-minute incubations and remeasurements, mean IgM levels were reduced by 94.55% and mean IgG levels by 99.46%. These data confirmed that the antibodies measured in these assays are directed against the S and N proteins of SARS-CoV-2.

The specificity of the IgG assay for identifying 852 SARS-CoV-2 RNA negative outpatients was 97.40% when using IgG only; when used in combination with the IgM, the specificity was 96.00%. In 200 SARS-CoV-2 negative hospitalized patients, the specificity for diagnosing negative patients was 97.5% for the IgG assay alone and 96.5% for both IgM and IgG. These experiments were carried out at Diazyme Laboratories, and test materials were obtained from various reference laboratories.

At Boston Heart Diagnostics, for validation we documented that positive values for both chemiluminescence assays are ≥1.0 AU/mL, with linear and reproducible reportable ranges of 1.0–10.0 AU/mL for IgM and of 0.20–100.00 AU/mL for IgG. Linearity studies documented $r^2$ values of 0.991 for both IgM and for IgG for actual values versus target values, with within- and between-run coefficients of variation based on 20 analyses at 4 concentration levels of 4.00% and 2.51% for IgM positive (≥1.0 AU/mL) control samples and 2.50% and 2.10% for IgG positive (≥1.0 AU/mL) control samples, respectively. Both these assays have received FDA EUA approval. In SARS-CoV-2 RNA positive patients (n = 55), the sensitivity for detecting positive subjects for the IgG assay was 98.40% for those with symptoms ≥15 days; together with IgM it was 98.20% based on studies at Boston Heart Diagnostics.

## Neutralizing antibody chemiluminescence assay

The SARS-CoV-2 neutralizing antibody assay utilized was obtained from Diazyme Laboratories (catalog number DZ901A). This assay is a competitive chemiluminescence immunoassay based on the specific interaction between the SARS-CoV-2 spike protein receptor binding domain (RBD) and the human angiotensin-converting enzyme 2 receptor (hACE2) on the surface of host cells. The assay and its validation with a cell-based assay have been previously described [15]. In the absence of SARS-CoV-2 neutralizing antibodies, hACE2 and RBD form complexes that generate a high chemiluminescent signal (measured in RLU). In the presence of SARS-CoV-2 neutralizing antibodies originating from human serum or plasma, the interaction between hACE2 and RBD is compromised; and the chemiluminescent signal is reduced in a dose-dependent manner. The assay has been validated with a cell-based assay as previously

described [27]. The assay was documented to have no interfering substances and to be specific for SARS-CoV-2. The assay showed excellent correlation with the cell-based SARS-CoV-2 Reporter Neutralizing Antibody Assay. Serum samples (n = 33) with neutralizing antibody values ≥2.60 AU/mL all showed >98.0% inhibition of viral infection in cell-based assay validation studies. In our laboratory, this assay was found to have within- and between-run coefficients of variation of <4.0%, with a positive value being ≥1.0 AU/mL and a linear range up to 30 AU/mL. This assay has been submitted to FDA for EUA. The remainder of our data using this assay are described in the results section below.

### Inflammatory marker assays

Serum hs-CRP and ferritin were measured using FDA-approved assays from Roche Diagnostics (Indianapolis, IN) on a Roche c701 automated COBAS analyzer. The IL-6 immunoassay was also obtained from Roche Diagnostics and was run on a Roche c801 automated COBAS analyzer. This assay has received FDA EUA for use in hospitalized COVID-19 patients (n = 49) who are at a >4-fold increased risk of needing a ventilator if their serum IL-6 values are >35 pg/mL versus patients with values ≤35 pg/mL. This information was provided in the Roche assay package insert. All assays had coefficients of variation of ≤4.0%.

### Statistical analysis

All statistical analyses were performed using R software, version 3.6 (R Foundation, Vienna, Austria). Categorical variables were expressed as frequencies and percentages, while continuous variables were expressed as median values with interquartile ranges (IQR, $25^{th}$–$75^{th}$ percentile values). The statistical significance of differences between groups were assessed using non-parametric Kruskal-Wallis analysis. Spearman correlation analyses were performed to assess interrelations of biochemical variables. Univariate and stepwise multivariate regression analyses were carried out to assess for the statistical significance of associations.

## Results

### Studies in RT-PCR RNA positive outpatients and inpatients

Data on serum inflammatory markers IL-6, hsCRP, and ferritin, and SARS-CoV-2 IgG, IgM, and neutralizing antibody levels in 100 SARS-CoV-2 RNA negative control subjects, 129 SARS-CoV-2 RNA positive outpatients, and 49 SARS-CoV-2 RNA positive inpatients are shown in Table 1. Median IL-6 levels were the same in controls and outpatients but were about 75-fold higher in inpatients as compared to other groups ($P$<0.0001). Median hs-CRP levels were very similar in control subjects and outpatients but were about 80-fold higher in inpatients as compared to other groups ($P$<0.0001). Similarly, median ferritin levels were similar in controls and outpatients but were about 9-fold higher in inpatients as compared to other groups ($P$<0.0001). Levels of inflammatory markers were only significantly elevated in inpatients as compared to controls and outpatients.

All control subjects had negative antibody levels (<1.0 AU/mL). Median IgG levels were about 300-fold and 600-fold higher in outpatients and inpatients as compared to controls (both $P$<0.0001). The wide variation in IgG response in RT-PCR positive outpatients and inpatients is shown in Fig 1. IgG values ranged 1.03–200.0 AU/mL in outpatients and 0.05–169.5 AU/mL in inpatients. Median IgM levels were about 1.8-fold and 5-fold higher in outpatients and inpatients as compared to control subjects (both $P$<0.0001). IgM values ranged from 1.09–13.58 AU/mL in outpatients and from 0.46–18.82 AU/mL in inpatients. Median neutralizing antibody levels using the described assay were about 12-fold and 24-fold higher in

**Table 1. Antibody and inflammatory biomarker response in SARS-CoV-2 PCR positive outpatients and PCR positive inpatients vs PCR negative control subjects.**

| | PCR Negative Controls* (N = 100) | PCR Positive Outpatients (N = 129) | PCR Positive Inpatients (N = 49) | *P* Value for Trend[†] |
|---|---|---|---|---|
| SARS-CoV-2 IgG, AU/mL | 0.05 (0.05–0.05) | 12.20 (3.79–35.20) | 30.61 (3.51–75.02) | 3.48 x 10^{-40} |
| SARS-CoV-2 IgM, AU/mL | 0.43 (0.34–0.54) | 0.76 (0.51–1.33) | 2.16 (1.11–3.56) | 7.50 x 10^{-24} |
| Neutralizing antibody, AU/mL[‡] | 0.30 (0.20–0.40) | 3.03 (2.04–5.27) | 7.17 (4.00 – 8.86) | 4.08 x 10^{-39} |
| Interleukin-6, pg/mL | 0.75 (0.75–2.66) | 0.75 (0.75–2.90) | 56.80 (28.49–482.60) | 1.71 x 10^{-24} |
| hs-CRP, mg/L | 0.83 (0.41–2.95) | 1.20 (0.40–2.70) | 66.90 (34.76–100.70) | 6.60 x 10^{-21} |
| Ferritin, ng/mL | 141.1 (79.6–248.4) | 144.2 (77.90–239.5) | 1311.0 (538.0–2035.0) | 6.99 x 10^{-18} |

Data are expressed as median (25th-75th percentile). Values that were outside the linear range of the assay were converted as follows: IgG <0.20 AU/mL to 0.05 AU/mL; IL-6 <1.5 to 0.75; IL-6 >5000 to 5500.

*Control subjects tested SARS-CoV-2 RNA not detected and SARS-CoV-2 IgG <0.2 AU/mL.

[†]*P* value for trend across the 3 subject groups.

AU, arbitrary units; hs-CRP, high sensitivity C reactive protein.

outpatients and inpatients, respectively, as compared to controls (both *P*<0.0001). Neutralizing antibody values ranged from 1.09–13.58 AU/mL in outpatients and from 0.35–18.82 AU/mL in inpatients. All median antibody levels were significantly higher in RT-PCR RNA positive patients than controls.

Correlations between inflammatory markers and antibody levels for the 100 controls subjects and the 178 positive outpatients and inpatients are shown in Table 2. IgG levels were strongly correlated with both neutralizing antibody levels as well as IgM levels, while IL-6 was most strongly correlated with hs-CRP values.

We sought to develop a multi-parameter algorithm to distinguish RT-PCR RNA positive subjects who required hospitalization from positive subjects not requiring hospitalization. The results of multivariate stepwise regression analysis for the prediction of need for hospitalization among RNA positive patients using cut-point analysis are shown in Table 3. Using the cutpoint of IL-6 ≥10 pg/mL, the odds ratio for hospitalization was 78.0, while for hs-CRP > 10 mg/L the odds ratio was > 58 (both highly significant). In hospitalized positive subjects, 97.7% had elevated levels for both parameters, while in positive outpatients this finding was only observed in 1.8%. The odds ratio for requiring hospitalization with elevated values of both parameters was >3,000 (C statistic 0.976, *P*<1.00 x 10^{-12}). Neither ferritin or IgG, IgM, or neutralizing antibody values added significant information to hospitalization risk prediction once IL-6 and hs-CRP were entered into the model.

## Studies with healthcare providers

Of 388 outpatients that had antibody testing in a healthcare provider's office (MMG) in the Riverdale area of the Bronx, NY, 17.5% had positive IgG values with or without positive IgM values, while another 4.9% had borderline IgG values between 0.50–1.0 AU/mL. Of these latter subjects, 60.0% had been or were symptomatic. Of 10 subjects in the borderline category, 3 had been previously RT-PCR RNA positive on NP swab testing, and 6 had a history of definite exposure. This healthcare provider felt that IgG values between 0.50–1.0 AU/mL should be classified as borderline. His data justified this conclusion.

Of 154 outpatients in Manhattan and New Jersey that had NP swabs and antibodies assessed, 85.8% were negative for any evidence of SARS-CoV-2. The remaining 14.2% (n = 22) were positive; of these subjects, 7 were carefully followed over time along with their family members, as well as 9 individual cases (total of 47 subjects). Many had the following symptoms: fever, chills, body aches, inability to sleep, fatigue, dry cough, loss of smell and taste,

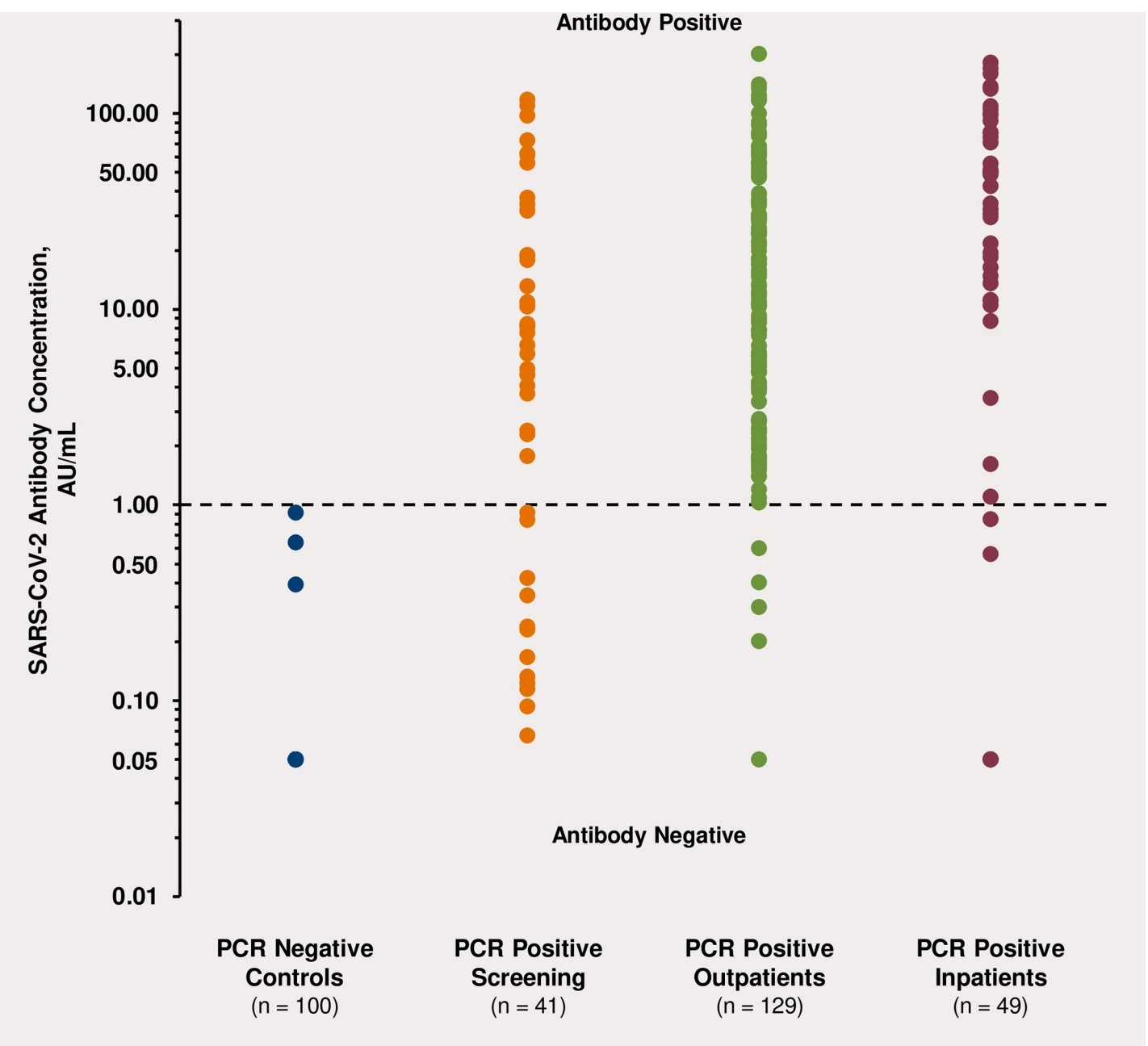

**Fig 1. Variability in SARS-CoV-2 IgG antibody response.** SARS-CoV-2 antibody response is shown in negative control subjects, for most of whom IgG values were <0.05 AU/mL and 100% were <1.0 AU/mL (dark blue circles); meat packing plant employees having antibody screening who were SARS-CoV-2 PCR RNA positive 2 weeks prior to testing (24.4% had IgG values <1.0 AU/mL) (orange circles); positive outpatients 4–6 weeks after positive SARS-CoV-2 RT-PCR RNA testing (3.9% had IgG values <1.0 AU/mL) (green circles); and positive SARS-CoV-2 RT-PCR RNA inpatients (6.1% had IgG values <1.0 AU/mL) (dark red circles). Dotted line indicates negative and positive SARS-CoV-2 IgG levels. IgG, immunoglobulin G; RT-PCR, reverse transcriptase-polymerase chain reaction.

shortness of breath, and diarrhea. Three cases (all aged >80 years) had to be hospitalized, and two required being placed on ventilators, with one of these latter patients dying. The data generated in these latter studies, based largely on the observations of one of our investigators (FC), indicated that 1) antibody testing was valuable for finding additional cases in family studies (observed in all families); 2) patients can have positive RNA results for up to 6 weeks (observed

**Table 2. Spearman correlation coefficient matrix analysis of antibody and inflammatory marker response in all subjects (N = 278).**

| | SARS-CoV-2 IgG | SARS-CoV-2 IgM | Neutralizing Antibodies | Interleukin-6 | hs-CRP | Ferritin |
|---|---|---|---|---|---|---|
| SARS-CoV-2 IgG | 1.000 | 0.642 ($<1.00 \times 10^{-12}$) | 0.872 ($<1.00 \times 10^{-12}$) | 0.287 ($1.37 \times 10^{-6}$) | 0.272 ($5.71 \times 10^{-6}$) | 0.260 ($1.46 \times 10^{-5}$) |
| SARS-CoV-2 IgM | 0.642 ($<1.00 \times 10^{-12}$) | 1.000 | 0.646 ($<1.00 \times 10^{-12}$) | 0.349 ($3.04 \times 10^{-9}$) | 0.320 ($7.77 \times 10^{-8}$) | 0.372 ($2.65 \times 10^{-10}$) |
| Neutralizing antibodies | 0.872 ($<1.00 \times 10^{-12}$) | 0.646 ($<1.00 \times 10^{-12}$) | 1.000 | 0.331 ($2.14 \times 10^{-8}$) | 0.340 ($1.07 \times 10^{-8}$) | 0.297 ($7.32 \times 10^{-7}$) |
| Interleukin-6 | 0.287 ($1.37 \times 10^{-6}$) | 0.349 ($3.04 \times 10^{-9}$) | 0.331 ($2.14 \times 10^{-8}$) | 1.000 | 0.743 ($<1.00 \times 10^{-12}$) | 0.409 ($2.49 \times 10^{-12}$) |
| hs-CRP | 0.272 ($5.71 \times 10^{-6}$) | 0.320 ($7.77 \times 10^{-8}$) | 0.340 ($1.07 \times 10^{-8}$) | 0.743 ($<1.00 \times 10^{-12}$) | 1.000 | 0.412 ($1.79 \times 10^{-12}$) |
| Ferritin | 0.260 ($1.46 \times 10^{-5}$) | 0.372 ($2.65 \times 10^{-10}$) | 0.297 ($7.32 \times 10^{-7}$) | 0.409 ($2.49 \times 10^{-12}$) | 0.412 ($1.79 \times 10^{-12}$) | 1.000 |

Data expressed as Spearman correlation coefficient r (p value).

hs-CRP, high sensitivity C-reactive protein; IgG, immunoglobulin G; IgM, immunoglobulin M

in 5 cases); and 3) patients with persistent symptoms often have persistently elevated IgM levels (observed in 11 cases).

In a separate analysis by one of our healthcare providers (JJ) of 217 employees at a local meat processing plant in Massachusetts tested with NP swabs, 24.0% were RT-PCR RNA positive. When 41 of these 52 positive subjects were retested in a screening study 2 weeks later, 73.2% still had positive NP swabs, 70.7% had positive IgG values, 9.8% had positive IgM values, and 63.4% had been symptomatic. Median IgG and IgM in all 41 subjects tested were 20.53 AU/mL and 0.54 AU/mL, respectively. As shown in Fig 1, there was a very large variability in their IgG response (range <0.20–117.7 AU/mL). In addition, there were 25 subjects that had prior RT-PCR RNA negative swab testing but requested antibody testing because of having significant symptoms and known exposure to subjects that had tested positive with RT- PCR RNA testing. Of these, 64.0% had positive IgG levels and 28.0% had positive IgM values, with all subjects in the latter group having persistent symptoms. Median IgG and IgM values in these positive subjects were 24.73 AU/mL and 1.31 AU/mL, respectively.

## Antibody testing in a reference laboratory population

Table 4 shows the results of serum antibody testing at Boston Heart Diagnostics between April 6[th] and December 1[st], 2020 by state in which more than 100 results were reported. The highest IgG and IgM positive rates were seen in meat packing plant employees in Nebraska (n = 352) with 19.0% having positive IgG values and 15.3% having positive IgM values. New York State had a fairly low positive rate because most subjects were sampled as part of health screening. In contrast, high IgG and IgM positive rates were observed in Pennsylvania from a program that screened newly symptomatic patients.

**Table 3. Prediction of need for hospitalization among SARS-CoV-2 RNA positive subjects.**

| | Odds Ratio[*] (5[th]-95[th] percentile CI) | P Value |
|---|---|---|
| Interleukin-6 (IL-6) ≥10 pg/mL | 78.0 (6.0–2001.9) | $1.33 \times 10^{-3}$ |
| hs-CRP ≥10 mg/L | 58.4 (7.6–1220.4) | $5.41 \times 10^{-4}$ |
| Both Parameters Elevated | 3861.0 (389.1–14,197.0) | $<1.00 \times 10^{-12}$ |

97.7% of positive subjects that met two or more of the above criteria required hospitalization, compared with 1.8% of positive subjects not requiring hospitalization, C statistic or area under the curve 0.976, *P*<0.0001).

[*]Odds ratio was determined by univariate and multivariate stepwise regression cut-point analysis. The addition of antibody and/or ferritin data did not add significantly to the odds ratio or the C statistic for the prediction of the need for hospitalization.

CI, confidence interval; hs-CRP, high sensitivity C reactive protein; IgM, immunoglobulin M

**Table 4. SARS-CoV-2 antibody testing by states with >100 tests.**

| State | Antibody Tests Done, N | SARS-CoV-2 IgG, % Positive | SARS-CoV-2 IgM, % Positive |
|---|---|---|---|
| Alabama | 151 | 6.62 | 1.32 |
| Arkansas | 323 | 7.43 | 3.72 |
| Arizona | 144 | 5.56 | 2.08 |
| California | 2,898 | 3.21 | 1.62 |
| Colorado | 270 | 4.44 | 3.33 |
| Connecticut | 1,398 | 7.65 | 2.43 |
| Florida | 1,448 | 8.7 | 3.66 |
| Georgia | 1,114 | 8.08 | 3.5 |
| Iowa | 141 | 2.13 | 3.55 |
| Idaho | 228 | 3.07 | 0.88 |
| Indiana | 832 | 13.58 | 5.77 |
| Massachusetts | 867 | 11.3 | 3.58 |
| Michigan | 1,150 | 7.57 | 3.83 |
| Missouri | 258 | 13.95 | 4.26 |
| North Carolina | 510 | 4.12 | 2.35 |
| Nebraska[*] | 363 | 20.94 | 15.98 |
| New Jersey | 361 | 12.19 | 6.37 |
| Nevada | 201 | 3.48 | 1.99 |
| New York[†] | 63,435 | 4.63 | 1.85 |
| Ohio | 142 | 1.41 | 2.11 |
| Oklahoma | 274 | 18.61 | 5.47 |
| Oregon | 1,473 | 3.87 | 3.19 |
| Pennsylvania[‡] | 180 | 10.56 | 7.22 |
| South Carolina | 103 | 5.83 | 1.94 |
| Texas | 1,610 | 7.27 | 4.16 |
| Virginia | 174 | 5.75 | 2.87 |
| Washington | 1,111 | 5.22 | 2.52 |

[*]Meat packing plant

[†]Mainly health screening

[‡]Newly symptomatic screening program.

Table 5 shows IgG antibody results in 81,624 subjects with values being positive (≥1.0 AU/mL) in 5.21%. In antibody positive subjects, median IgG levels increased progressively and very significantly by age group. Median values in subjects were 3.14 AU/mL if <30 years of age, 4.38 AU/mL if 30–44 years of age, 7.89 AU/mL if 45–54 years of age, 9.52 AU/mL if 55–64 years of age, and 10.64 AU/mL if ≥65 years of age. Very similar trends were seen in both females and males, as well as for the percentage of positive subjects having values >20 AU/mL. No clear age or gender trends were observed for the percentage of subjects having positive IgG values. Moreover, gender differences were much less pronounced than age differences with regard to median positive IgG levels. Table 5 also shows data for IgM values, which were measured in a subset of 61,126 subjects. Of these subjects, 2.96% had positive values of ≥1.0 AU/mL. While median values for IgM only increased modestly by age group, the percentage of subjects with positive values was significantly greater in older subjects than younger subjects. This finding was especially true in females, going from 2.11% in the youngest group to 3.46% in the oldest group.

**Table 5. SARS-CoV-2 antibody levels by age and gender[*].**

| | Age <30 Years (N = 15,595; 19.1%) | Age 30–44 Years (N = 19,967; 24.4%) | Age 45–54 Years (N = 15,757; 19.3%) | Age 55–64 Years (N = 16,866; 20.6%) | Age ≥65 Years (N = 13,572; 16.6%) | % Difference, Older vs Younger |
|---|---|---|---|---|---|---|
| **IgG ≥1.0 AU/mL** | | | | | | |
| Total positive, N (%) | 828 (5.32%) | 919 (4.61%)[‡1] | 820 (5.21%) | 949 (5.63%) | 738 (5.44%) | +2.26 |
| Median value (IQR) | 3.14 (1.68–7.4) | 4.38 (1.85–12.31)[‡2] | 7.89 (2.15–27.85)[‡3] | 9.52 (2.78–33.17)[‡4] | 10.46 (2.69–39.98)[‡5] | +233.12 |
| Female positive subjects, N (%) | 469 (5.13%) | 479 (4.16%)[‡6] | 450 (4.87%) | 493 (5.02%) | 387 (5.20%) | +1.36 |
| Median value (IQR), AU/mL | 3.10 (1.72–6.94) | 4.50 (1.89–11.9)[‡7] | 6.40 (1.92–22.23)[‡8] | 9.02 (2.82–25.93)[‡9] | 10.44 (3.38–34.43)[‡10] | +236.77 |
| IgG >20 AU/mL, N (%) | 37 (0.40%) | 78 (0.68%)[‡11] | 118 (1.28%)[‡12] | 163 (1.66%)[‡13] | 147 (1.97%)[‡14] | +392.5 |
| Male positive subjects, N (%)[†1] | 359 (5.59%) | 440 (5.24%) | 370 (5.69%) | 456 (6.48%)[‡15] | 351 (5.73%) | +2.5 |
| Median value (IQR), AU/mL | 3.25 (1.62–8.28) | 4.16 (1.8–13.05)[‡16] | 10.97 (2.76–39.13)[‡17†2] | 10.18 (2.73–39.28)[‡18] | 10.64 (2.04–46.95)[‡19] | +227.38 |
| IgG >20 AU/mL, N (%) | 40 (0.62%) | 75 (0.89%) | 139 (2.14%)[‡20] | 180 (2.56%)[‡21] | 139 (2.27%)[‡22] | +266.13 |
| **IgM ≥1.0 AU/mL** | | | | | | |
| Total positive, N (%) | 250 (2.41%) | 390 (2.68%) | 356 (2.96%)[‡23] | 440 (3.31%)[‡24] | 373 (3.42%)[‡25] | +41.91 |
| Median values (IQR), AU/mL | 1.38 (1.14–1.99) | 1.45 (1.14–2.28) | 1.61 (1.22–2.9)[‡26] | 1.65 (1.23–2.56)[‡27] | 1.55 (1.23–2.55)[‡28] | +12.32 |
| Female positive subjects, N (%) | 131 (2.11%) | 202 (2.33%) | 180 (2.49%) | 195 (2.48%) | 209 (3.46%)[‡29] | +63.98 |
| Median values (IQR), AU/mL | 1.32 (1.13–1.79) | 1.43 (1.14–2.24) | 1.48 (1.17–2.67)[‡30] | 1.52 (1.21–2.39)[‡31] | 1.47 (1.2–2.39)[‡32] | +11.36 |
| Male positive subjects, N (%)[†3] | 119 (2.87%) | 188 (3.20%) | 176 (3.68%)[‡33] | 245 (4.51%)[‡34] | 164 (3.36%) | +17.07 |
| Median values (IQR), AU/mL | 1.48 (1.15–2.08)[†4] | 1.48 (1.15–2.34) | 1.75 (1.28–3.14)[‡35] | 1.71 (1.23–2.7) | 1.66 (1.27–2.82) | +12.16 |

[*] A total of 150,222 serum samples were submitted to our laboratory for antibody measurements, and this number decreased to 83,153 samples when only the first sample was utilized, and this value decreased to 81,624 after removing those subjects without age or gender information. Their median age was 48.0 years (IQR 30–55), and they were 57.77% female. A subset of 61,126 subjects (median age 50.0 years [IQR 35–61]; 58.89% female) also had IgM values measured. Of all subjects, 89.1% had an IgG value <0.20 AU/mL; 3.72% had an IgG value 0.20–<0.50 AU/mL; 1.97% had an IgG value 0.50–<1.0 AU/mL; 3.84% had an IgG value 1.0–20.0 AU/mL; and 1.37% had an IgG value >20.0 AU/mL. For IgM, 97.04% had a value <1.0 AU/mL; 2.88% had a value of 1.0–10.0 AU/mL; and 0.08% had a value >10.0 AU/mL. The Spearman correlation coefficient between IgG and IgM for all subjects with values >1.0 AU/mL was r = 0.39 (P < 0.001).

[†] For males of all ages had IgG and IgM values compared with their female counterparts. For these comparisons [†1]P = 1.11 x $10^{-8}$; [†2]P = 2.37 x $10^{-5}$; [†3]P = 7.25 x $10^{-13}$; [†4]P = 4.41 x $10^{-3}$.

[‡] For age comparisons to <30-year age group. The percentage values represent a comparison between the age ≥65-year group and the <30-year age group. [‡1]P = 2.39 x $10^{-5}$; [‡2]P = 1.31 x $10^{-6}$; [‡3]P = 1.24 x $10^{-25}$; [‡4]P = 3.92 x $10^{-41}$; [‡5]P = 2.96 x $10^{-38}$; [‡6]P = 9.7 x $10^{-5}$; [‡7]P = 4.37 x $10^{-5}$; [‡8]P = 2.60 x $10^{-9}$; [‡9]P = 1.13 x $10^{-22}$; [‡10]P = 1.17 x $10^{-27}$; [‡11]P = 1.12 x $10^{-5}$; [‡12]P = 1.69 x $10^{-10}$; [‡13]P = 5.08 x $10^{-17}$; [‡14]P = 1.67 x $10^{-21}$; [‡15]P = 3.38 x $10^{-5}$; [‡16]P = 4.66 x $10^{-5}$; [‡17]P = 8.43 x $10^{-5}$; [‡18]P = 1.79 x $10^{-7}$; [‡19]P = 9.64 x $10^{-5}$; [‡20]P = 3.06 x $10^{-13}$; [‡21]P = 1.73 x $10^{-18}$; [‡22]P = 1.41 x $10^{-14}$; [‡23]P = 1.25 x $10^{-5}$; [‡24]P = 5.29 x $10^{-5}$; [‡25]P = 1.7 x $10^{-5}$; [‡26]P = 2.1 x $10^{-5}$; [‡27]P = 2.71 x $10^{-5}$; [‡28]P = 1.0 x $10^{-4}$; [‡29]P = 6.61 x $10^{-6}$; [‡30]P = 6.8 x $10^{-5}$; [‡31]P = 1.34 x $10^{-5}$; [‡32]P = 6.51 x $10^{-5}$; [‡33]P = 3.63 x $10^{-6}$; [‡34]P = 3.63 x $10^{-5}$; [‡35]P = 1.6 x $10^{-5}$.

# Discussion

An initial goal of our studies was to examine the relationships of inflammation markers and antibody levels in SARS-CoV-2 positive patients requiring hospitalization, as compared to such subjects not requiring hospitalization, in order to develop a risk algorithm for need for hospitalization. The highest median inflammatory marker hs-CRP, IL-6, and ferritin levels

and the highest median IgG, IgM, and neutralizing antibody levels were noted in hospitalized COVID-19 patients. We also noted a high degree of variability in IgG response as shown in Fig 1. The inflammatory markers are part of the criteria for so called "cytokine storm" associated with an exaggerated immune response along with markedly elevated blood levels of white blood cells associated with a high COVID-19 mortality [28–31]. In a meta-analysis, IL-6 levels were reported to be >12-fold elevated in COVID-19 related respiratory distress [30]. Moreover, serum levels of IL-6 >80 pg/mL and hs-CRP >97 mg/L have been reported to identify correctly 80% of hospitalized COVID-19 patients requiring a ventilator with C statistic values of 0.90 and 0.97, respectively [31]. The Infectious Diseases Society of America has recommended that the criteria for systemic inflammation in COVID-19 patients be a CRP value of ≥75 mg/L, and that such patients be given both dexamethasone and monoclonal antibody therapy [32]. However, inflammatory marker criteria for hospitalization for COVID-19 have not been adequately addressed.

In our multivariate analysis, only two parameters allowed for the very precise prediction of the need for hospitalization in RT-PCR RNA positive patients, namely, having combined elevations of IL-6 ≥10.0 pg/mL and hs-CRP ≥10 mg/L. Surprisingly, once these parameters were in the prediction model, neither ferritin or antibody levels added significant information about hospitalization risk. In our data set, having hs-CRP value >10 mg/L alone increased hospitalization risk 58-fold, while also having IL-6 ≥10 pg/mL increased hospitalization risk >3000-fold in COVID positive patients with a highly significant C statistic value of 0.976. Therefore, using these serum markers, one can very accurately predict need for hospitalization among SARS-CoV-2 RT-PCR RNA positive patients.

Another goal of our studies was to investigate the interrelationships of IgG, IgM, neutralizing antibodies and inflammatory markers. We noted that IgG levels were most strongly correlated with both neutralizing antibody levels and IgM levels, while IL-6 was most strongly correlated with hs-CRP values, consistent with prior studies [21]. A great advantage of the serum or plasma neutralizing assay we used in our studies was its ease of use on high throughput automated instruments and its reproducibility. Moreover, the results of this assay were found to be very highly correlated with results obtained using a cell-based assay [15, 27].

Another goal of our studies was to assess the clinical utility of antibody assays in case finding. We documented that antibody testing was valuable to identify cases and to ascertain potential exposure and level of immunity. SARS-CoV-2 RNA detection using PCR methodology may not always be optimal in exposed subjects because of inadequate sample collection by NP or nasal swabs, or after several weeks the virus may no longer be present in the nasal cavities. The advantage of antibody testing is that IgG levels usually persist for many months after SARS-CoV-2 infection. We have also noted a high degree of variability in IgG antibody response in RNA positive patients. Laboratories that only report a positive or negative value do not detect this large variability. Moreover, only about 50% of RNA positive outpatients had IgG levels >6.5 AU/mL, sufficient to provide estimated antibody titers of >1:320 as per FDA guidance, and only about one-third had plasma IgG levels >20 AU/mL, sufficient to provide estimated antibody titers >1:1000 for potential plasma donation [16–20]. In this regard, monoclonal antibody therapy would appear to be preferable because of the known amount of antibody being provided.

In our individual and cluster studies, we have noted that antibody testing allows for the identification of exposed individuals, especially in those that were negative based on NP swab testing, usually ≥4 weeks following infection. Most of these family cluster and individual cases studies were carried out by one of the co-authors (FC). She justifiably emphasized the value of both RNA and antibody testing in her practice. Her data clearly documented the benefits of semi-quantitative IgG and IgM testing for case finding in family clusters and exposed subjects

who were RNA negative. Her data also indicated that RNA swabs can remain positive for up to 6 weeks, even though such patients may no longer be able to infect other people [33, 34]. In her cluster and case data, we also clearly observed that long-term elevated IgM levels were often associated with persistent illness and symptoms. At the present time, very few healthcare providers are measuring COVID-19 antibody levels; instead, there has been a frenzy of nasal swab RNA testing [3–6]. Unfortunately, such testing in the United States has often been accompanied by a lack of public health measures as well as contact tracing to combat the spread of COVID-19. In our view, antibody testing provides an excellent measure of prior exposure and potential immunity that has been greatly under-utilized in the United States [35].

Another goal of our studies was to assess the effects of age on serum SARS-CoV-2 IgG and IgM antibody levels. In a large number of outpatients with potential SARS-CoV-2 exposure, about 5% had positive IgG values and about 3% had positive IgM values. It has been reported by the Centers for Disease Control and Prevention (CDC) that serum SARS-CoV-2 antibody levels were positive in 1.0–6.5% of 16,025 subjects in various parts of the United States, suggesting that infection rates were 6–24 times higher than reported at that time [36]. These percentages are similar to our data. Based on CDC data, over 95% of deaths from COVID occur in the >45-year age group, even though about 70% of the cases occur in those <45 years of age. The ≥65 years of age category accounts for ~10% of all SARS-CoV-2 cases and ~80% of SARS-CoV-2 mortality [26]. In our studies in a population of over 80,000 subjects, median IgG levels were more than 3-fold higher in those ≥65 years as compared to those <30 years of age. Possibly older subjects with positive antibody levels mount a greater IgG response in order to compensate for the decreased overall cellular immunity found in the elderly as compared to the young [37, 38].

## Conclusions

Our data are consistent with the following conclusions: 1) serum SARS-CoV-2 IgG antibody levels are significantly correlated with neutralizing antibody levels; 2) having both IL-6 ≥10 pg/mL and hs-CRP ≥10.0 mg/L very accurately predicts the need for hospitalization in COVID-19 positive patients; 3) elevated SARS-CoV-2 IgG level measurements are useful in identifying cases in exposed subjects and family clusters, 4) elevated SARS-CoV-2 IgM levels are often associated with persistent COVID-19 symptoms and disease; and 5) SARS-CoV-2 IgG antibody levels are significantly higher in positive older subjects than in younger positive subjects.

## Supporting information

**S1 Dataset.**
(XLSX)

**S2 Dataset.**
(XLSX)

## Acknowledgments

We thank the laboratory staff at Boston Heart Diagnostics, Framingham, MA, and Diazyme Laboratories, Poway, CA, and the clinical staff at the Comite Center for Precision Medicine and Health, New York, NY, St. Francis Hospital/Trinity Health of New England, Hartford, CT, Atkinson Family Practice, Amherst, MA, Grajower Clinical Practice, the Bronx, NY, and

the Advanced Cardiology Institute, Fort Lee, NJ for their efforts and commitment to SARS-CoV-2 testing, diagnosis, and treatment.

## Author Contributions

**Conceptualization:** Ernst J. Schaefer, Latha Dulipsingh, Florence Comite, Jessica Jimison, Martin M. Grajower, Nathan E. Lebowitz, Andrew S. Geller, Margaret R. Diffenderfer, Lihong He, Ben Saida, Chong Yuan.

**Data curation:** Ernst J. Schaefer, Latha Dulipsingh, Florence Comite, Jessica Jimison, Martin M. Grajower, Nathan E. Lebowitz, Maxine Lang, Andrew S. Geller, Margaret R. Diffenderfer, Lihong He, Gary Breton, Michael L. Dansinger, Ben Saida, Chong Yuan.

**Formal analysis:** Ernst J. Schaefer, Maxine Lang, Andrew S. Geller, Margaret R. Diffenderfer, Lihong He, Ben Saida, Chong Yuan.

**Funding acquisition:** Ernst J. Schaefer, Latha Dulipsingh, Florence Comite, Jessica Jimison, Martin M. Grajower, Nathan E. Lebowitz.

**Investigation:** Ernst J. Schaefer, Latha Dulipsingh, Florence Comite, Jessica Jimison, Martin M. Grajower, Nathan E. Lebowitz, Maxine Lang, Andrew S. Geller, Margaret R. Diffenderfer, Lihong He, Gary Breton, Michael L. Dansinger, Ben Saida, Chong Yuan.

**Methodology:** Ernst J. Schaefer, Latha Dulipsingh, Florence Comite, Jessica Jimison, Martin M. Grajower, Nathan E. Lebowitz, Maxine Lang, Andrew S. Geller, Margaret R. Diffenderfer, Lihong He, Gary Breton, Michael L. Dansinger, Ben Saida, Chong Yuan.

**Project administration:** Ernst J. Schaefer, Latha Dulipsingh, Martin M. Grajower, Nathan E. Lebowitz, Maxine Lang, Andrew S. Geller, Margaret R. Diffenderfer.

**Resources:** Ernst J. Schaefer, Latha Dulipsingh, Florence Comite, Jessica Jimison, Martin M. Grajower, Nathan E. Lebowitz.

**Supervision:** Ernst J. Schaefer, Latha Dulipsingh, Florence Comite, Jessica Jimison, Martin M. Grajower, Nathan E. Lebowitz, Maxine Lang, Andrew S. Geller.

**Validation:** Ernst J. Schaefer, Andrew S. Geller, Margaret R. Diffenderfer, Lihong He, Gary Breton, Ben Saida, Chong Yuan.

**Visualization:** Ernst J. Schaefer, Maxine Lang, Margaret R. Diffenderfer.

**Writing – original draft:** Ernst J. Schaefer.

**Writing – review & editing:** Ernst J. Schaefer, Latha Dulipsingh, Florence Comite, Jessica Jimison, Martin M. Grajower, Nathan E. Lebowitz, Maxine Lang, Andrew S. Geller, Margaret R. Diffenderfer, Lihong He, Gary Breton, Michael L. Dansinger, Ben Saida, Chong Yuan.

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
