## [Decision Letter · Decision Letter 0]

26 Feb 2021

PONE-D-21-03508

Clinical utility of Corona Virus Disease-19 serum IgG, IgM, and neutralizing antibodies and inflammatory markers

PLOS ONE

Dear Dr. Schaefer,

Thank you for submitting your manuscript to PLOS ONE. After careful consideration, we feel that it has merit but does not fully meet PLOS ONE’s publication criteria as it currently stands. Therefore, we invite you to submit a revised version of the manuscript that addresses the points raised during the review process. Please take in account the questions and the numerous improvements suggested by the two reviewers. Please provide detailled indications of how you answer in the rebutal letter as well as where are the addition or modification in the revised version (line and page).

We look forward to receiving your revised manuscript.

Kind regards,

Pierre Roques, Ph.D.

Academic Editor

PLOS ONE

Journal Requirements:

2. Please include your actual numerical p-values in Table 2.

3. Please provide all names and catalog numbers of all assays used in your experiments. For modified assays, please briefly describe what modifications were in place.

4. Please provide the catalog numbers, source, and dilutions of all antibodies used in this study.

5. Please provide information on the source of the coronavirus strains HKU1, NL63, OC43, or 229E. Please ensure that you state whether researchers obtained personal information related to these samples or whether samples were de0identified before researchers obtained them.

6. We note that you have indicated that data from this study are available upon request. PLOS only allows data to be available upon request if there are legal or ethical restrictions on sharing data publicly. For information on unacceptable data access restrictions, please see http://journals.plos.org/plosone/s/data-availability#loc-unacceptable-data-access-restrictions.

7. Thank you for providing the following Funding Statement: 

'Support for this research was provided by Boston Heart Diagnostics, Framingham, MA. EJS, ML, ASG, MRD, LH, GB, and MLD were either full-time or part-time employees of Boston Heart Diagnostics when the research was conducted. The funders had no role in the study design, data collection and analysis, decision to publish, or preparation of the manuscript. '

We note that one or more of the authors are employed by a commercial company Diazyme Laboratories, Inc.  We also note that one or more of the authors is affiliated with the funding organization, Boston Heart Diagnostics indicating the funder may have had some role in the design, data collection, analysis or preparation of your manuscript for publication; in other words, the funder played an indirect role through the participation of the co-authors.

a. Please provide an amended Funding Statement declaring these commercial affiliations.

If the funding organization did not play a role in the study design, data collection and analysis, decision to publish, or preparation of the manuscript and only provided financial support in the form of authors' salaries and/or research materials, please review your statements relating to the author contributions, and ensure you have specifically and accurately indicated the role(s) that these authors had in your study in the Author Contributions section of the online submission form. Please make any necessary amendments directly within this section of the online submission form. 

Please also update your Funding Statement to include the following statement: “The funder provided support in the form of salaries for authors [insert relevant initials], but did not have any additional role in the study design, data collection and analysis, decision to publish, or preparation of the manuscript. The specific roles of these authors are articulated in the ‘author contributions’ section.”

If the funding organization did have an additional role, please state and explain that role within your Funding Statement.

b. Please also provide an updated Competing Interests Statement declaring these commercial affiliations along with any other relevant declarations relating to employment, consultancy, patents, products in development, or marketed products, etc.  

Within your Competing Interests Statement, please confirm that this commercial affiliation does not alter your adherence to all PLOS ONE policies on sharing data and materials by including the following statement: "This does not alter our adherence to  PLOS ONE policies on sharing data and materials.” (as detailed online in our guide for authors http://journals.plos.org/plosone/s/competing-interests). If this adherence statement is not accurate and  there are restrictions on sharing of data and/or materials, please state these.

Please note that we cannot proceed with consideration of your article until this information has been declared.

8. PLOS requires an ORCID iD for the corresponding author in Editorial Manager on papers submitted after December 6th, 2016. Please ensure that you have an ORCID iD and that it is validated in Editorial Manager. To do this, go to ‘Update my Information’ (in the upper left-hand corner of the main menu), and click on the Fetch/Validate link next to the ORCID field. This will take you to the ORCID site and allow you to create a new iD or authenticate a pre-existing iD in Editorial Manager. Please see the following video for instructions on linking an ORCID iD to your Editorial Manager account: https://www.youtube.com/watch?v=_xcclfuvtxQ

Reviewers' comments:

Reviewer's Responses to Questions

**Comments to the Author**

1. Is the manuscript technically sound, and do the data support the conclusions?

Reviewer #1: Yes

Reviewer #2: Yes

2. Has the statistical analysis been performed appropriately and rigorously? 

Reviewer #1: Yes

Reviewer #2: Yes

3. Have the authors made all data underlying the findings in their manuscript fully available?

Reviewer #1: Yes

Reviewer #2: Yes

4. Is the manuscript presented in an intelligible fashion and written in standard English?

Reviewer #1: Yes

Reviewer #2: No

5. Review Comments to the Author

Reviewer #1: Schaefer et al. present a serology study comprising nearly 80,000 serum samples using a commercially available and validated SARS-CoV-2 IgG/IgM chemiluminescence assay. The authors further characterise a subset of samples with known diagnostic PCR results, and validate their results using orthogonal competitive bindings assays as a surrogate for viral neutralisation as well as IL-6, CRP, and ferritin levels. The cohort and data are impressive, and the authors use case studies to highlight the utility of antibody testing. While many of the conclusions and correlations have been described, the large number of samples presented here, using a harmonised detection system, makes this manuscript worthy of publication.

I have a few comments that should be addressed prior to to publication:

1. One of the central findings of the paper is the difference is seropositivity rates and antibody titre (as measured by AU) by age. The data is currently presented as a table, but I think this should be additionally visualised in figure form with age on the x axis and AU levels and seropositivity levels on the y axes. Given the large number of samples in each age group, it would be useful to see the age distribution of these values with smaller bin sizes.

2. For analyses of PCR-confirmed subjects, the authors should present antibody/IL-6/CRP/ferritin levels in relation to time after a positive test or symptoms, if this information is available.

3. Fig 1 shows the range of IgG antibody titres in two separate cohorts. The data is presented as change from an assumed baseline value of 0.05 AU, which is unnecessary. Plotting the range of AU values with a dashed line at 0.05 to indicate the positivity threshold is a clearer and more accurate way to show this data.

4. The authors present a case report of 388 outpatients from a healthcare provider, and describe a subset with borderline positive IgG values. This discussion is worth expanding as this would be of great interest to the clinical community. Of these borderline subjects, were they IgM negative as well? Can the authors go back and test these serum samples for CRP, IL-6, and ferritin as with other samples? If the authors can re-test these samples and provide a diagnostic differentiator for samples with borderline IgG levels, it would add greatly to the study and be of clinical interest.

Reviewer #2: The information in this manuscript is important. The body of work is significant. As presented, though, the manuscript is not easy to read: (a) Some parts of the manuscript are not presented in a standard format; in particular, information that should be in the Results section are given in the Methods section, (b) Some sentences are incomplete, and (c) Insufficient detail or explanation is presented for some of the statements. Some points for the authors to consider:

ABSTRACT

The study design is not clear, and this can be easily remedied by adding some essential details.

L4: Suggest mentioning here the 3 serum inflammatory markers that were measured.

L7: 79,005 of what type of subjects from when?

L8: Please define the context you sue for the term "level" (ie, concentration, activity, etc).

L9: Median what type of IgG? Neutralizing? IgG1, IgG2, etc?

L10: “SARS-CoV-2 positive RNA” comes out of nowhere. The authors assume the reader knows this is from a diagnostic RT-PCR test. But RT-PCR tests can continue to be positive even though infectious virus is not formed in people recovering from COVID (as pointed out by the authors in the DISCUSSION section).

L12: IMPORTANT: The authors have not defined “case”. One reason there is confusion regarding COVID-19 statistics is that the word ‘case’ is defined as some to be clinically apparent illness for which there is a lab confirmed test for SARS-CoV-2, others define ‘case’ to mean a positive SARS-CoV-2 test, whether or not the person develops illness, etc.

L15: the authors claim the antibodies are ‘neutralizing’ based on work performed by others. This is misleading. Maybe a better descriptor would be to first mention that a ‘surrogate’ test for neutralizing antibody was used.

L-19: “possibly to compensate for decreased cellular immunity”. This is speculation, as this was not measured in the study. Suggest leaving that out of the abstract and including that in the discussion.

INTRODUCTION

L25-26: The first sentence is awkward. The subject is COVID-19, yet the sentence ends with “has caused a world-wide pandemic”. Moreover, a pandemic is generally world-wide, so that wording is redundant. The word “infection” has different connotations in various disciplines. For example, about 80% of people who were infected with Zika virus (including some who could transmit it sexually) did not know they harbored the virus. The point is infection in virology just means the host harbors the virus, regardless of whether an apparent illness ensues. But in medicine, the term typically refers to an apparent illness. Maybe state the first sentence something like “Severe acute respiratory syndrome coronavirus 2 (SARS-CoV-2) is the causative agent of the Coronavirus disease 2019 (COVID-19) pandemic. A COVID-19 diagnosis is typically confirmed by RT-PCR detection of SARS-CoV-2 RNA in…..specimens collected within five…”.

L31 It is not clear what the authors mean by “…case finding in family clusters”. Do they mean finding evidence of SARS-CoV-2 infection among a family unit, some who developed and others that did not develop symptoms?

L32 Plasma or serum? Or both?

L34: Should be obvious to most readers but for sake of clarity, please write as “In SARS-CoV-2 RNA-positive subjects”.

L35 – 36: Please clarify: Do the authors mean to state that IgM antibodies are not virus neutralizing? Both IgM and IgA should contribute to virus neutralization.

L37 Presumably, similar results using either method for the items discussed in 34 to 37.

L40-41: Why mention fingerstick testing? Suggest completing the thought by adding additional verbiage for bringing this up.

L44: NOTE: In L32, plasma is mentioned, but a stated goal is to find out what is in serum, and the reader presumes that the antibody levels will be the same and that the serology assays are best performed with serum than plasma. Is this a correct presumption? It would be helpful if the authors added additional explanation.

L45: the authors state that antibodies are detectable around 5 days post development of symptoms. So what does ‘symptom prediction’ mean? To predict symptoms that will arise (ie, to ‘predict’), or to correlate Ab findings with recorded signs/symptoms?

L48: What type of “risk”? Risk for developing ….?

Materials and Methods

L56: More than 100 what kind of results?

L80: The heading is not quite right. RT-PCR is used to detect RNA, may or may not be in a virion. It is also redundant to say “…CoV-2 viral ..” . The authors might instead write this heading something like: Detection of SARS-CoV-2 RNA by RT-PCR” or “RT-PCR Detection of SARS-CoV-2 RNA”.

L87: Please make it easier for the reader by briefly stating what your modification was. You could write something like “…..[4]. Briefly,……”.

L89: So what? Note also that the reference is a preprint article. Suggest the authors mention that the Viracor test has been shown to be valid by….and has a sensitivity and specificity of……

L92: As previously described by the authors or others?

L96: What type of anti-human antibodies?

L102 and 103: Are the authors referring to the results they obtained? If so, move to the results section.

L104 to 111. It is not clear how these tests were performed. What were the antigens for those tests? Obtained from where? In lines 104 – 105, the authors state the samples tested positive for antibodies to…..are they saying SOME tested positive or they ALL tested positive for all the antigens?

L107 - 108: Italicize Mycoplasma pneumoniae; do not capitalize pneumoniae. Italicize C. pneumoniae and C. albicans.

L112 – 116: What was the volume of the serum? What were the final concentrations of serum and protein?

L117 to 122 should be in RESULTS section.

L123 to 128: The authors do not specify what these are in refence to; the current study?

129: This section has a mixture of methods and results.

RESULTS

L162: What is a reference laboratory population? Samples obtained from a reference laboratory?

L173: 79,005 of what type of subjects? Specify here. Samples collected when?

DISCUSSION

L307-308: Please comment: So what if there is variability? Wouldn’t that be expected considering the subjects different past or ongoing health histories, the antibody levels are measured on different days post-onset of symptoms, and the virus strains that affect these people may differ in virulence?

L313-326: The authors should consider commenting on the following: for RT-PCR tests, it is acknowledged that sample collection itself can be problematic (ie, a negative test can be from a poorly collected sample).

L332: REF 26 is for the “cell-based assay”; who exactly showed that the assay used by the authors demonstrated equivalence?

6. PLOS authors have the option to publish the peer review history of their article (what does this mean?). If published, this will include your full peer review and any attached files.

Reviewer #1: No

Reviewer #2: No

---

## [Author Response · Author response to Decision Letter 0]

21 Apr 2021

Rebuttal Letter, Responses to Editor, and Responses to Reviewers

PONE-D-21-03508

Clinical utility of Corona Virus Disease-19 serum IgG, IgM, and neutralizing antibodies and inflammatory markers

PLOS ONE

Response: Please note title change: now: “Inflammatory Markers, Corona Virus Disease-19 Serology, Hospitalizations, Case Finding, and Aging

Dear Dr. Schaefer,

Thank you for submitting your manuscript to PLOS ONE. After careful consideration, we feel that it has merit but does not fully meet PLOS ONE’s publication criteria as it currently stands. Therefore, we invite you to submit a revised version of the manuscript that addresses the points raised during the review process. Please take in account the questions and the numerous improvements suggested by the two reviewers. Please provide detailed indications of how you answer in the rebuttal letter as well as where are the addition or modification in the revised version (line and page).

Response: We have provided this information. Please see below as well as see the covering letter. This document corresponds to responses to the editor and the reviewers. 

Updated Financial Disclosures and Conflicts of Interest as well as Author Contributions to the Research: 

Overall Funding Statement: 

This research was funded by Boston Heart Diagnostics, Framingham, MA, St. Francis Hospital, Trinity Health of New England, Hartford, Ct, Comite Center for Precision Medicine and Health, New York, NY, Atkinson Family Practice, Amherst, MA, Advanced Cardiology Institute, Fort Lee, NJ, and Diazyme Laboratories, Poway, CA. The authors were employees of Boston Heart Diagnostics (EJS, ML, ASG, MRD, LH, GB, MLD, LH, GB, MLD), Trinity Health of New New England (LD) Comite Center for Precision Medicine and Health (FC), Atkinson Family Practice (JJ), Grajower Medical Practice (MMG), Advanced Cardiology Institute (NEL), and Diazyme Laboratories (BS, CY). The funders provided support in the form of salaries for authors but did not have any additional role in the study design, data collection and analysis, decision to publish, or preparation of the manuscript. The specific roles of these authors are articulated in the ‘author contributions’ section. These commercial affiliation does not alter adherence to PLOS ONE policies on sharing data and materials. The conclusions expressed are solely those of the authors.

“Author Contributions” 

Ernst J Schaefer: Conceptualization, Data Curation, Formal Analysis, Funding Acquisition, Investigation, Methodology, Project Administration, Resources, Supervision, Validation, Visualization, Writing Original and Editing. 

Latha Dulipsingh: Conceptualization, Data Curation, Funding Acquisition, Investigation, Methodology, Project Administration, Resources, Supervision, Editing 

Florence Comite: Conceptualization, Data Curation, Funding Acquisition, Investigation, Methodology, Resources, Supervision, Editing

Jessica Jimison: Conceptualization, Data Curation, Funding Acquisition, Investigation, Methodology, Resources, Supervision, Editing

Martin M. Grajower: Conceptualization, Data Curation, Funding Acquisition, Investigation, Methodology, Project Administration, Resources, Supervision, Editing

Nathan E. Lebowitz: Conceptualization, Data Curation, Funding Acquisition, Investigation, Methodology, Project Administration, Resources, Supervision, Editing

Maxine Lang: Data Curation, Formal Analysis, Investigation, Methodology, Project Administration, Supervision, Validation, Visualization, Editing. 

Andrew S. Geller: Conceptualization, Data Curation, Investigation, Methodology, Project Administration, Supervision, Validation, Editing. 

Margaret R. Diffenderfer: Conceptualization, Data Curation, Formal Analysis, Investigation, Methodology, Project Administration, Validation, Visualization, Editing. 

Lihong He: Conceptualization, Data Curation, Formal Analysis, Investigation, Methodology, Validation, Editing. 

Gary Breton: Data Curation, Investigation, Methodology, Validation, Editing. 

Michael L. Dansinger: Data Curation, Investigation, Methodology, Editing

Ben Saida: Conceptualization, Data Curation, Formal Analysis, Investigation, Methodology, Validation, Editing. 

Chong Yuan: Conceptualization, Data Curation, Formal Analysis, Investigation, Methodology, Validation, Editing. 

Response: All information about our protocols is provided in the manuscript. 

We look forward to receiving your revised manuscript.

Response: Please see attached – thank you. 

Kind regards,

Pierre Roques, Ph.D.

Academic Editor

PLOS ONE

4/6/21

Response: 

Dear Dr. Roques, 

Please see above responses to your requests. Thank you for your letter and getting the paper reviewed. We are now resubmitting the paper. We have done our best to respond to the reviewers’ comments and criticisms and have made the requested changes requiring a whole new data analysis with different age cutpoints. Point by point responses to the reviewers can be found below. A marked copy of the manuscript is attached, as is a clean copy. In addition, we have attended to all the details outlined above and below. Thank you. 

Sincerely yours, 

Ernst J. Schaefer, MD 

Chief Medical Officer & Laboratory Director 

Boston Heart Diagnostics

Framingham, MA 

Journal Requirements:

Response: We have now done so. 

2. Please include your actual numerical p-values in Table 2.

Response: We have now done so. 

3. Please provide all names and catalog numbers of all assays used in your experiments. For modified assays, please briefly describe what modifications were in place.

Response: We have now done so. 

4. Please provide the catalog numbers, source, and dilutions of all antibodies used in this study.

Response: We have now done so. 

5. Please provide information on the source of the coronavirus strains HKU1, NL63, OC43, or 229E. Please ensure that you state whether researchers obtained personal information related to these samples or whether samples were deidentified before researchers obtained them.

Response: We have now done so. 

6. We note that you have indicated that data from this study are available upon request. PLOS only allows data to be available upon request if there are legal or ethical restrictions on sharing data publicly. For information on unacceptable data access restrictions, please see http://journals.plos.org/plosone/s/data-availability#loc-unacceptable-data-access-restrictions.

Response: We have now uploaded all the data. 

Response: We have now provided the requested information. There are no restrictions. 

If there are no restrictions, please upload the minimal anonymized data set necessary to replicate your study findings as either Supporting Information files or to a stable, public repository and provide us with the relevant URLs, DOIs, or accession numbers. Please see http://www.bmj.com/content/340/bmj.c181.long for guidelines on how to de-identify and prepare clinical data for publication. For a list of acceptable repositories, please see http://journals.plos.org/plosone/s/data-availability#loc-recommended-repositories.

Response: We have now uploaded the anonymized data as requested. 

7. Thank you for providing the following Funding Statement: 

'Support for this research was provided by Boston Heart Diagnostics, Framingham, MA. EJS, ML, ASG, MRD, LH, GB, and MLD were either full-time or part-time employees of Boston Heart Diagnostics when the research was conducted. The funders had no role in the study design, data collection and analysis, decision to publish, or preparation of the manuscript. '

We note that one or more of the authors are employed by a commercial company Diazyme Laboratories, Inc. We also note that one or more of the authors is affiliated with the funding organization, Boston Heart Diagnostics indicating the funder may have had some role in the design, data collection, analysis or preparation of your manuscript for publication; in other words, the funder played an indirect role through the participation of the co-authors.

Please provide an amended Funding Statement declaring these commercial affiliations.

Response: We have now done so. 

If the funding organization did not play a role in the study design, data collection and analysis, decision to publish, or preparation of the manuscript and only provided financial support in the form of authors' salaries and/or research materials, please review your statements relating to the author contributions, and ensure you have specifically and accurately indicated the role(s) that these authors had in your study in the Author Contributions section of the online submission form. Please make any necessary amendments directly within this section of the online submission form. 

Please also update your Funding Statement to include the following statement: “The funder provided support in the form of salaries for authors [insert relevant initials], but did not have any additional role in the study design, data collection and analysis, decision to publish, or preparation of the manuscript. The specific roles of these authors are articulated in the ‘author contributions’ section.”

Response: We have now done so. Please see above. 

If the funding organization did have an additional role, please state and explain that role within your Funding Statement.

Please also provide an updated Competing Interests Statement declaring these commercial affiliations along with any other relevant declarations relating to employment, consultancy, patents, products in development, or marketed products, etc. 

Response: We have now done so. 

Within your Competing Interests Statement, please confirm that this commercial affiliation does not alter your adherence to all PLOS ONE policies on sharing data and materials by including the following statement: "This does not alter our adherence to PLOS ONE policies on sharing data and materials.” (as detailed online in our guide for authors http://journals.plos.org/plosone/s/competing-interests). If this adherence statement is not accurate and there are restrictions on sharing of data and/or materials, please state these.

Response: We have now provided the requested information. 

Please note that we cannot proceed with consideration of your article until this information has been declared.

Response: We have now provided the requested information. 

Response: We have now provided this statement as requested. 

8. PLOS requires an ORCID iD for the corresponding author in Editorial Manager on papers submitted after December 6th, 2016. Please ensure that you have an ORCID iD and that it is validated in Editorial Manager. To do this, go to ‘Update my Information’ (in the upper left-hand corner of the main menu), and click on the Fetch/Validate link next to the ORCID field. This will take you to the ORCID site and allow you to create a new iD or authenticate a pre-existing iD in Editorial Manager. Please see the following video for instructions on linking an ORCID iD to your Editorial Manager account: https://www.youtube.com/watch?v=_xcclfuvtxQ

Response: The ORCID iD for the corresponding author (Dr. Schaefer) is: 000-0002-7158-3085. 

Responses to the Reviewers’ Comments

Reviewers' comments:

Reviewer's Responses to Questions

Comments to the Author

1. Is the manuscript technically sound, and do the data support the conclusions?

Reviewer #1: Yes

Reviewer #2: Yes

2. Has the statistical analysis been performed appropriately and rigorously? 

Reviewer #1: Yes

Reviewer #2: Yes

3. Have the authors made all data underlying the findings in their manuscript fully available?

Reviewer #1: Yes

Reviewer #2: Yes

Response: We have now uploaded all primary anonymized data as requested. 

4. Is the manuscript presented in an intelligible fashion and written in standard English?

Reviewer #1: Yes

Reviewer #2: No

Response: We have now modified the manuscript to improve clarity, correctness, and remove any ambiguity in light of the response of Reviewer #2. 

5. Review Comments to the Author

Reviewer #1: Schaefer et al. present a serology study comprising nearly 80,000 serum samples using a commercially available and validated SARS-CoV-2 IgG/IgM chemiluminescence assay. The authors further characterise a subset of samples with known diagnostic PCR results, and validate their results using orthogonal competitive bindings assays as a surrogate for viral neutralization as well as IL-6, CRP, and ferritin levels. The cohort and data are impressive, and the authors use case studies to highlight the utility of antibody testing. While many of the conclusions and correlations have been described, the large number of samples presented here, using a harmonized detection system, makes this manuscript worthy of publication.

Response: We have not been able to identify a prior manuscript that clearly documents significantly higher SARS-CoV-2 IgG and IgM serum antibody levels in the elderly as compared to the young in a large population. We do believe that the information we have provided is novel. 

I have a few comments that should be addressed prior to publication:

1. One of the central findings of the paper is the difference is seropositivity rates and antibody titre (as measured by AU) by age. The data is currently presented as a table, but I think this should be additionally visualised in figure form with age on the x axis and AU levels and seropositivity levels on the y axes. Given the large number of samples in each age group, it would be useful to see the age distribution of these values with smaller bin sizes.

Response: We did try to present a figure of the data, but we feel that the data is best presented as a table. We have redone the analysis and have increased the number of age groups from 3 (i.e <45, 45-64, and >65 years of age) to 5; i.e <30, 30-44, 45-54, 55-64, and >65 years of age). Please see new Table 5, with at least 500 positive subjects per age group. 

2. For analyses of PCR-confirmed subjects, the authors should present antibody/IL-6/CRP/ferritin levels in relation to time after a positive test or symptoms, if this information is available.

Response: Unfortunately, our data mainly represents one time point after a positive test. In the large data set, we did not have the time point information. However, we did have it for the studies in hospitalized and non-hospitalized positive subjects as well as in the results from individual healthcare providers and have now presented that data first to respond to these issues. 

3. Fig 1 shows the range of IgG antibody titres in two separate cohorts. The data is presented as change from an assumed baseline value of 0.05 AU, which is unnecessary. Plotting the range of AU values with a dashed line at 0.05 to indicate the positivity threshold is a clearer and more accurate way to show this data.

Response: We have now modified the figure to include IgG values in control subjects, subjects participating in a screening study, positive outpatients, and positive inpatients. We have plotted the individual IgG values and in the text we now indicate the time from diagnosis based on a positive nasal swab for SARS-CoV-2 RNA. 

4. The authors present a case report of 388 outpatients from a healthcare provider, and describe a subset with borderline positive IgG values. This discussion is worth expanding as this would be of great interest to the clinical community. Of these borderline subjects, were they IgM negative as well? Can the authors go back and test these serum samples for CRP, IL-6, and ferritin as with other samples? If the authors can re-test these samples and provide a diagnostic differentiator for samples with borderline IgG levels, it would add greatly to the study and be of clinical interest.

Response: Unfortunately, those samples were no longer available for further analysis, unlike the samples from the subjects studied as part of the research protocol at St. Francis Hospital. 

Reviewer #2: The information in this manuscript is important. The body of work is significant. As presented, though, the manuscript is not easy to read: (a) Some parts of the manuscript are not presented in a standard format; in particular, information that should be in the Results section are given in the Methods section, (b) Some sentences are incomplete, and (c) Insufficient detail or explanation is presented for some of the statements. Some points for the authors to consider:

Response: We agree and have made multiple changes in this regard. 

ABSTRACT

The study design is not clear, and this can be easily remedied by adding some essential details.

L4: Suggest mentioning here the 3 serum inflammatory markers that were measured. Response: We have made this change. 

L7: 79,005 of what type of subjects from when? Response: We have clarified this issue in the methods section. 

L8: Please define the context you use for the term "level" (ie, concentration, activity, etc). Response: We have clarified this issue, also see methods. 

L9: Median what type of IgG? Neutralizing? IgG1, IgG2, etc? Response: We have now clarified this issue in the methods section. 

L10: “SARS-CoV-2 positive RNA” comes out of nowhere. The authors assume the reader knows this is from a diagnostic RT-PCR test. But RT-PCR tests can continue to be positive even though infectious virus is not formed in people recovering from COVID (as pointed out by the authors in the DISCUSSION section). Response: We have now clarified this issue in the abstract as well as in the methods section. 

L12: IMPORTANT: The authors have not defined “case”. One reason there is confusion regarding COVID-19 statistics is that the word ‘case’ is defined as some to be clinically apparent illness for which there is a lab confirmed test for SARS-CoV-2, others define ‘case’ to mean a positive SARS-CoV-2 test, whether or not the person develops illness, etc. Response: We have now done our best to define “case” as anyone who tests positive by RT PCR. 

L15: the authors claim the antibodies are ‘neutralizing’ based on work performed by others. This is misleading. Maybe a better descriptor would be to first mention that a ‘surrogate’ test for neutralizing antibody was used. Response: We have now provided more detail in the methods section about validation of the neutralization antibody test. 

L-19: “possibly to compensate for decreased cellular immunity”. This is speculation, as this was not measured in the study. Suggest leaving that out of the abstract and including that in the discussion. Response: We have made the recommended deletion in the abstract. 

INTRODUCTION

L25-26: The first sentence is awkward. The subject is COVID-19, yet the sentence ends with “has caused a world-wide pandemic”. Moreover, a pandemic is generally world-wide, so that wording is redundant. The word “infection” has different connotations in various disciplines. For example, about 80% of people who were infected with Zika virus (including some who could transmit it sexually) did not know they harbored the virus. The point is infection in virology just means the host harbors the virus, regardless of whether an apparent illness ensues. But in medicine, the term typically refers to an apparent illness. Maybe state the first sentence something like “Severe acute respiratory syndrome coronavirus 2 (SARS-CoV-2) is the causative agent of the Coronavirus disease 2019 (COVID-19) pandemic. A COVID-19 diagnosis is typically confirmed by RT-PCR detection of SARS-CoV-2 RNA in…..specimens collected within five…”. Response: We have made the recommended changes. 

L31 It is not clear what the authors mean by “…case finding in family clusters”. Do they mean finding evidence of SARS-CoV-2 infection among a family unit, some who developed and others that did not develop symptoms? 

Response: We mean finding people with positive antibody levels who never had RT PCR testing or had negative RT PCR testing previously. 

L32 Plasma or serum? Or both? Response: Serum, although the assays work well for either type of sample. 

L34: Should be obvious to most readers but for sake of clarity, please write as “In SARS-CoV-2 RNA-positive subjects”. Response: We have now made this change. 

L35 – 36: Please clarify: Do the authors mean to state that IgM antibodies are not virus neutralizing? Both IgM and IgA should contribute to virus neutralization. 

Response: We agree and have made this change. The correlation between IgG and neutralizing antibody levels was stronger than between neutralizing antibodies and IgM levels; although both correlations were statistically highly significant. 

L37 Presumably, similar results using either method for the items discussed in 34 to 37. 

L40-41: Why mention fingerstick testing? Suggest completing the thought by adding additional verbiage for bringing this up. Response: We have deleted any mention of fingerstick testing. 

L44: NOTE: In L32, plasma is mentioned, but a stated goal is to find out what is in serum, and the reader presumes that the antibody levels will be the same and that the serology assays are best performed with serum than plasma. Is this a correct presumption? It would be helpful if the authors added additional explanation. Response: We have indicated that serum is what we have used. However, all our antibody assays have been validated for both serum and plasma. 

L45: the authors state that antibodies are detectable around 5 days post development of symptoms. So what does ‘symptom prediction’ mean? To predict symptoms that will arise (ie, to ‘predict’), or to correlate Ab findings with recorded signs/symptoms?

L48: What type of “risk”? Risk for developing ….? Response: We have clarified this issue to indicate that about 5 days after symptoms develop, IgM antibodies may first be detected in those who develop symptoms. 

Materials and Methods

L56: More than 100 what kind of results? Response: At least 100 positive antibody levels. 

to 111. It is not clear how these tests were performed. What were the antigens for those tests? Obtained from where? In lines 104 – 105, the authors state the samples tested positive for antibodies to…..are they saying SOME tested positive or they ALL tested positive for all the Response: These were serum analyses for antibodies, not antigen tests. This is now clarified in the text. 

L107 - 108: Italicize Mycoplasma pneumoniae; do not capitalize pneumoniae. Italicize C. pneumoniae and C. albicans. Response: We have done so as requested. 

L112 – 116: What was the volume of the serum? What were the final concentrations of serum and protein? Response: The antibody assays were run on chemiluminescence analyzers and they require a minimum volume of 150 microliters of serum. The results are based on relative light units which correlate with arbitrary units/mL or AU/mL as described in the methods section. 

L117 to 122 should be in RESULTS section.

L123 to 128: The authors do not specify what these are in reference to; the current study?

129: This section has a mixture of methods and results. Response: We have left information in the methods section provided by the manufacturer as well as our own validation studies as required for laboratory developed tests. However, everything else has been moved to the Results section. 

RESULTS

L162: What is a reference laboratory population? Samples obtained from a reference laboratory?

L173: 79,005 of what type of subjects? Specify here. Samples collected when?

Response: Boston Heart Diagnostics is a service or reference laboratory approved and certified by CLIA and CAP. The samples we receive are sent to us by overnight FedEx on ice packs by healthcare providers who want specific tests run on their clients or patients. Specifically, for the large population they requested Diazyme IgM and IgG antibody levels (both FDA EUA approved). We ran the Diazyme neutralizing antibodies during development. It is now offered as a lab developed test. The inflammatory markers hs-CRP, IL-6, and ferritin (FDA approved tests) were run on subjects participating in Dr. Dulipsingh’s protocol. We have clarified these points in the paper and have provided much more information about the study populations and the exclusions (see methods section). 

DISCUSSION

L307-308: Please comment: so what if there is variability? Wouldn’t that be expected considering the subjects different past or ongoing health histories, the antibody levels are measured on different days post-onset of symptoms, and the virus strains that affect these people may differ in virulence? Response: This is a good point. We are merely documenting the variability. 

L313-326: The authors should consider commenting on the following: for RT-PCR tests, it is acknowledged that sample collection itself can be problematic (ie, a negative test can be from a poorly collected sample). Response: We agree that this is a potential problem; however, our paper focuses on antibody and inflammatory markers and not RT-PCR testing. However, we have added a statement about this issue in the discussion. 

L332: REF 26 is for the “cell-based assay”; who exactly showed that the assay used by the authors demonstrated equivalence? Response: We have now added additional information about the neutralizing antibody assay we used and its validation with a cell-based assay has been previously described (see reference 15). The cell-based assay has also been described (see reference 27). Serum samples were sent to that laboratory at the University of Texas, Galveston, for the validation studies. 

 6. PLOS authors have the option to publish the peer review history of their article (what does this mean?). If published, this will include your full peer review and any attached files. Response: No thank you. 

Do you want your identity to be public for this peer review? For information about this choice, including consent withdrawal, please see our Privacy Policy.

Reviewer #1: No

Reviewer #2: No

---

## [Decision Letter · Decision Letter 1]

24 May 2021

Corona Virus Disease-19 serology, inflammatory markers, hospitalizations, case finding, and aging

PONE-D-21-03508R1

Dear Dr. Schaefer,

We’re pleased to inform you that your manuscript has been judged scientifically suitable for publication and will be formally accepted for publication once it meets all outstanding technical requirements.

Kind regards,

Pierre Roques, Ph.D.

Academic Editor

PLOS ONE

Additional Editor Comments (optional):

Reviewers' comments:

Reviewer's Responses to Questions

**Comments to the Author**

1. If the authors have adequately addressed your comments raised in a previous round of review and you feel that this manuscript is now acceptable for publication, you may indicate that here to bypass the “Comments to the Author” section, enter your conflict of interest statement in the “Confidential to Editor” section, and submit your "Accept" recommendation.

Reviewer #1: All comments have been addressed

2. Is the manuscript technically sound, and do the data support the conclusions?

Reviewer #1: Yes

3. Has the statistical analysis been performed appropriately and rigorously? 

Reviewer #1: Yes

4. Have the authors made all data underlying the findings in their manuscript fully available?

Reviewer #1: Yes

5. Is the manuscript presented in an intelligible fashion and written in standard English?

Reviewer #1: Yes

6. Review Comments to the Author

Reviewer #1: The authors have sufficiently addressed all of my concerns, and I recommend publication. One minor point that should be addressed is that the title should read 'Coronavirus disease-19' rather than 'Corona virus disease-19'.

7. PLOS authors have the option to publish the peer review history of their article (what does this mean?). If published, this will include your full peer review and any attached files.

Reviewer #1: No

---

## [Editor Report · Acceptance letter]

3 Jun 2021

PONE-D-21-03508R1 

Corona Virus Disease-19 serology, inflammatory markers, hospitalizations, case finding, and aging 

Dear Dr. Schaefer:

I'm pleased to inform you that your manuscript has been deemed suitable for publication in PLOS ONE. Congratulations! Your manuscript is now with our production department. 

Kind regards, 

on behalf of

Dr. Pierre Roques 

Academic Editor

PLOS ONE